# IN-CONTEXT PRETRAINING: LANGUAGE MODELING BEYOND DOCUMENT BOUNDARIES

**Weijia Shi**[1,2]    **Sewon Min**[1,2]    **Maria Lomeli**[1]    **Chunting Zhou**[1]
**Margaret Li**[1,2]    **Gergely Szilvasy**[1]    **Rich James**[1]    **Xi Victoria Lin**[1]
**Noah A. Smith**[2,3]    **Luke Zettlemoyer**[1,2]    **Scott Yih**[1]    **Mike Lewis**[1]
[1]Meta AI    [2]University of Washington    [3] Allen Institute for AI
swj0419@cs.washington.edu

## ABSTRACT

Large language models (LMs) are currently trained to predict tokens given document prefixes, enabling them to directly perform long-form generation and prompting-style tasks which can be reduced to document completion. Existing pretraining pipelines train LMs by concatenating random sets of short documents to create input contexts but the prior documents provide no signal for predicting the next document. We instead present IN-CONTEXT PRETRAINING, a new approach where language models are pretrained on a sequence of *related* documents, thereby explicitly encouraging them to read and reason across document boundaries. We can do IN-CONTEXT PRETRAINING by simply changing the document ordering so that each context contains related documents, and directly applying existing pretraining pipelines. However, this document sorting problem is challenging. There are billions of documents and we would like the sort to maximize contextual similarity for every document without repeating any data. To do this, we introduce approximate algorithms for finding related documents with efficient nearest neighbor search and constructing coherent input contexts with a graph traversal algorithm. Our experiments show IN-CONTEXT PRETRAINING offers a simple and scalable approach to significantly enhance LMs' performance: we see notable improvements in tasks that require more complex contextual reasoning, including in-context learning (+8%), reading comprehension (+15%), faithfulness to previous contexts (+16%), long-context reasoning (+5%), and retrieval augmentation (+9%).

## 1    INTRODUCTION

Large language models (LMs) are trained to complete documents; each token is predicted given the context provided by the prefix of the document it appears in. Such contexts can be widely varied, especially at pretraining scale, allowing models to excel on diverse tasks such as instruction-following (Ouyang et al., 2022), conversational interfaces (OpenAI, 2023), reading comprehension (Zhang et al., 2020), and in-context learning (Brown et al., 2020). However, recent studies highlight that LMs sometimes struggle to understand more complex contexts: they can fail to follow instructions accurately (McKenzie et al., 2023; Efrat & Levy, 2020; Liu & Liu, 2023), struggle with reasoning over conditioned documents (Liu et al., 2023; Shi et al., 2023a), and exhibit high variance in in-context learning (Zhao et al., 2021). In this paper, we present IN-CONTEXT PRETRAINING, a new pretraining method that learns to predict tokens conditioned on a sequence of related documents, explicitly enabling the model to read and reason about much more varied and longer contexts that go beyond document boundaries.

Current LM training pipelines concatenate random sets of shorter documents to create longer context windows. However, the prior documents provide no signal for predicting the next document, incurring unnecessary computational overhead for tokens that do not require communication between them (de Vries, 2023). IN-CONTEXT PRETRAINING instead reorders the pretraining data by combining several semantically related documents to create a coherent input context, thereby exposing LMs to long *relevant* contexts and providing pretraining signals beyond document boundaries. We illustrate this via an example in Figure 1: when predicting the following tokens for the phrase "*For 2022, FIFA set the prize money at $42m,*" a previous document stating that the "*World Cup never awarded*

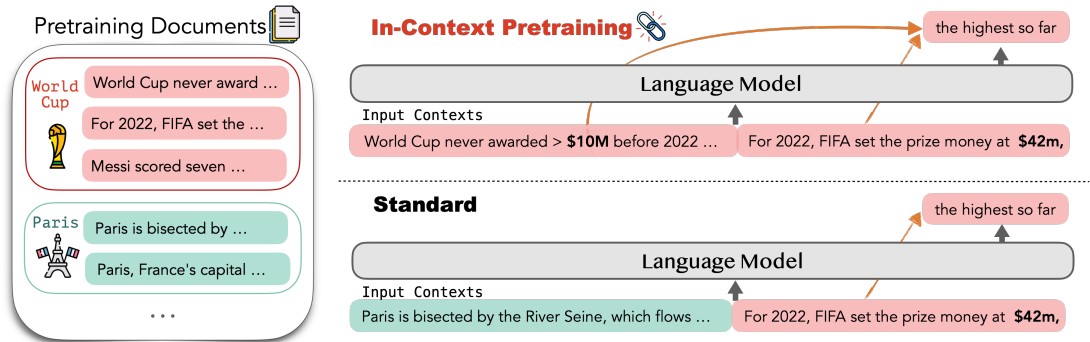

Figure 1: **Overview of IN-CONTEXT PRETRAINING**. Different from the *standard* pretraining strategy that place randomly shuffled documents in the input context, IN-CONTEXT PRETRAINING places related documents in the same context, making models learn to reason across prior documents. For example, when predicting the following tokens for the phrase "*For 2022, FIFA set the prize money at $42m,*" LMs could reference prior documents stating "*World Cup never awarded more than $10M before 2022*" and learn to infer that "*the highest so far.*"

*more than $10M before 2022*" could be in the context, enabling the prediction of a continuation like "*the highest so far.*" As IN-CONTEXT PRETRAINING only changes document ordering and leaves all other aspects of LM pretraining untouched, it can be easily integrated into existing pretraining pipelines for large-scale LMs.

However, this document sorting problem is challenging. LMs are typically trained on billions of documents and we would like to sort them to maximize document similarity in the input context windows without repeating any data. We introduce two new approximate algorithms to tackle these challenges. We use a retrieval model paired with an efficient search index to build a document graph that pairs each document with its nearest-neighbors based on its semantic similarity in the embeddings space. We also formulate document sorting as a travelling salesman problem, for which we develop an effective algorithm that maximizes similarity of documents to their context while also ensures that each document is included only once.

To evaluate the effectiveness of IN-CONTEXT PRETRAINING, we pretrain language models from 0.3 to 7 billion parameters on 300 billion tokens from the CommonCrawl dataset (Wenzek et al., 2020). Across all model scales, IN-CONTEXT PRETRAINING LMs (ICLM) demonstrate strong language modeling and downstream task performance, outperforming LMs pretrained using the standard approach on the same corpus. We observe various improvements resulting from IN-CONTEXT PRETRAINING compared with existing LMs: (1) **in-context learning** with an average increase of 8% across 8 datasets; (2) **reading comprehension**, with an average of 15% improvement on 8 reading comprehension tasks; (3) **outputs that are more faithful** to prior contexts (+16%); (4) **long context reasoning**, showing a 5% boost; and (5) **retrieval augmentation**, leading to 9% gains when augmenting with external knowledge such as documents retrieved from Wikipedia. Our results demonstrate that, by simply altering order of the pretraining documents, IN-CONTEXT PRETRAINING offers a scalable and simple approach to significantly enhance understanding and reasoning over their full contexts. Code are publicly released at github.com/swj0419/in-context-pretraining.

## 2 IN-CONTEXT PRETRAINING

The *standard* practice in pretraining is to form input contexts by concatenating random documents until reaching the maximum context length. It then trains the LM using a language modeling objective on the input contexts. However, training LMs on randomly concatenated documents does not offer additional learning signals compared with training on each document individually. In contrast, IN-CONTEXT PRETRAINING generates more coherent input contexts by concatenating semantically related documents together during pretraining. As depicted in Figure 2, IN-CONTEXT PRETRAINING consists of two steps: it first finds related documents at scale (§2.1) and then constructs input contexts using these related documents (§2.2). Successively, we use the contexts formed with

semantically related documents to pretrain LMs with a language modeling objective. Since IN-CONTEXT PRETRAINING is identical to existing pretraining recipes for LMs, except for changing how input contexts are built, it can be easily integrated into existing pretraining pipelines for large-scale LMs.

## 2.1 FINDING RELATED DOCUMENTS AT SCALE: RETRIEVING NEIGHBOR DOCUMENTS

To find related documents at scale, we link documents within the pretraining corpus $\mathcal{D}$ using a retrieval model. Specifically, for each document $d_i \in \mathcal{D}$, a dense retrieval model is used to retrieve the top-$k$ most similar documents, represented as $N(d_i)$. The retrieval model uses approximate nearest neighbours search for efficient pairwise similarity comparison between any two documents, making it scalable for finding related documents in web-scale pretraining corpora.

**Retrieval.**  Our retrieval process employs the contriever model (Izacard et al., 2022). This model maps each document $d_i \in \mathcal{D}$ to an embedding $\mathbf{E}(d_i)$ by taking the mean pooling of the last hidden representation over the tokens in $d_i$. The cosine similarity is then used to determine the similarity between any two documents:

$$s(d_i, d_j) = \cos(\mathbf{E}(d_i), \mathbf{E}(d_j)) \tag{1}$$

The retrieval model uses approximate nearest neighbour search with the faiss library (Johnson et al., 2019; Douze et al., 2024). We use product quantization (Jégou et al., 2011) to reduce the memory footprint and an IVF (inverted file) index structure to conduct efficient pairwise similarity search together with faiss big batch search. The OIVFBBS faiss framework is leveraged for this task, OIVFBBS refers to conducting offline search with queries of big batches with faiss inverted indexes. Further details can be found in Appendix  A.2 and in the OIVFBBS demo in the faiss github repository github.com/facebookresearch/faiss/tree/main/demos/offline_ivf.

During the retrieval process, when computing pairwise similarity among each document in the pretraining corpus, we found that the pretraining corpus contains many near duplicate documents. Hence, we further leverage the retrieval scores to eliminate near duplicate documents from the pretraining corpus. More details can be found in Appendix  A.1. In §4.2, we show that this deduplication step is crucial for achieving good performance of language models.

## 2.2 CREATING INPUT CONTEXTS: DOCUMENT GRAPH TRAVERSAL

Given a set of documents $\mathcal{D} = \{d_i\}$ and nearest neighbours for each document $N(d_i)$, our goal is to sort the documents to create input contexts such that each of them consists a list of *related* documents. Formally, we aim to form a set of input contexts $\mathcal{C}_1 \cdots \mathcal{C}_m$ where each context $\mathcal{C}_i = \{d_1, ...d_k\} \subset \mathcal{D}$ and $\bigcup_{i=1}^{m} \mathcal{C}_i = \mathcal{D}$. Ideally, documents in $\mathcal{C}_i$ are nearest neighbors of each others.

A straightforward approach to form $\mathcal{C}_1 \cdots \mathcal{C}_m$ is to directly place each document and its retrieved top-$k$ documents together in the same input context (referred to as $k$NN), which has been used in some retrieval-augmented pretraining methods (Guu et al., 2020; Levine et al., 2022). This $k$NN approach maintains document similarity within each context but creates the data repeating problem: some documents frequently appear as nearest neighbors of other documents, causing that different input contexts contain overlapping documents, i.e., $\exists i \neq j, \mathcal{C}_i \bigcap \mathcal{C}_j \neq \emptyset$. The data repeating problem exposes LMs to a less diverse set of documents given a fixed computational budget and could lead to overfitting of popular documents. Instead, we aim to build a set of contexts in a way that each document is included only once, which can be cast as a graph traversal problem.

**Algorithm 1** Maximum Traveling Salesman

**Input**: Document graph $\mathcal{G} = (\mathcal{D}, \mathcal{L})$
    $N(d_i)$ returns nearest neighbors for $d_i$
    $\texttt{min\_deg}(\mathcal{D})$ returns a min-degree doc
**Output**: A path $P$

1:  $P \leftarrow []$
2:  **while** $|\mathcal{D}| > 0$ **do**
3:      $d_i \leftarrow \texttt{min\_deg}(\mathcal{D})$
4:      $P.append(d_i)$
5:      $\mathcal{D}.remove(d_i)$
6:      **while** $N(d_i) \cap \mathcal{D} \neq \emptyset$ **do**
7:          $d_j \leftarrow \arg\min_{d \in N(d_i) \cap \mathcal{D}} \text{sim}(d_i, d)$
8:          $d_i \leftarrow d_j$
9:          $P.append(d_i)$
10:         $\mathcal{D}.remove(d_i)$
11:     **end while**
12: **end while**
13: **return**  $P$

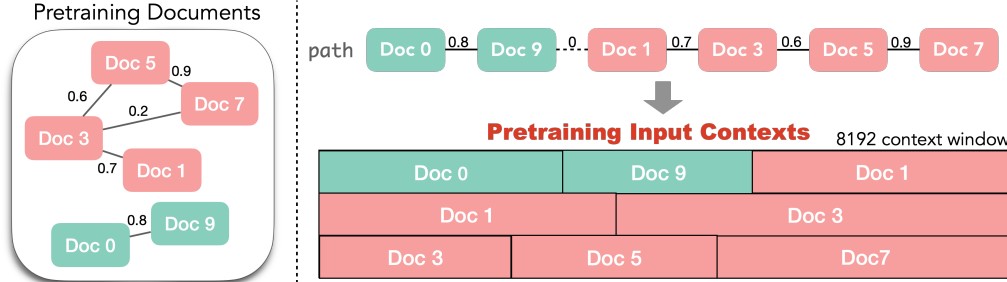

Figure 2: **Illustration of IN-CONTEXT PRETRAINING**. IN-CONTEXT PRETRAINING first finds related documents at scale to create a document graph (§2.1) and then builds pretraining input contexts by traversing the document graph (§2.2). Along the path, documents are concatenated into a sequence and subsequently divided to form fixed-sized input contexts (e.g., 8192 token length).

**Document graph traversal.**    To achieve our goal of maximizing the chance that the related documents are concatenated together, an intuitive approach is to find a single path that visits each document once and maximize the chance that related documents are visited sequentially. Then we subsequently segment the path into multiple input contexts. We formulate it as the *maximum traveling salesman problem* (Flood, 1956) that aims to find the maximum weight path that traverses all nodes exactly once. We represent each document as a node in the graph and use document similarity as a edge weight. We design an undirected weighted graph representing the documents, symbolized as $\mathcal{G} = (\mathcal{D}, \mathcal{L})$. Here, $\mathcal{D}$ represents the set of documents, while $(d, d^*) \in \mathcal{L}$ is a edge if $d^* \in N(d_i)$ or $d_i \in N(d^*)$. The weight of each edge corresponds to the document similarity (Equation 1).

Solving large traveling salesman problems exactly is NP hard, but greedy algorithms are known to provide an efficient approximate solution. We adopt this approach, introducing modifications to better suit our context. Algorithm 1 shows the method to construct a maximum weight path. We show a path identified by our algorithm in Figure 2. Our algorithm starts by selecting a yet-to-be-visited document with the minimum degree as the starting node (Doc 0). The algorithm then progressively extends the current path by navigating to its unvisited neighboring document with highest weight (Doc 9), adding the document node to the path. This process continues until the path reaches a node where all neighboring documents have been visited, which happens because our graph is not complete, and only contains edges between documents where one is within the other's $k$ nearest neighbors. In this case, we extend the graph with an edge of weight 0 to a random unvisited *minimum degree* document (Doc 1), and continue the above process. The motivation for starting at minimum degree documents is that they are most likely to have all their neighbors visited first, and therefore be connected to dissimilar documents in the final path.

As a final step, we traverse the documents along the path and concatenate them to create fixed-sized input contexts suitable for pretraining. It is important to note that when forming the input training batches, we ensure the diversity among different input contexts within the same batch.

## 3    EXPERIMENTS

In this section, we describe details of our pretraining setup (§3.1), the baseline methods we use for comparison (§3.2), and experimental results (§3.3).

### 3.1    PRETRAINING SETUP

Since IN-CONTEXT PRETRAINING leaves other details of model training unchanged, and only changes the document ordering so that each context contains related documents, we can directly integrate it into pretraining pipelines as a preprocessing step during batching. For our experiment, we adopt the model architecture and pretraining objective of LLaMA (Touvron et al., 2023a;b) and pretrain LMs from scratch.

**Pretraining Datasets.** We use the English Commoncrawl dataset (Wenzek et al., 2020), the widely-used data source for pretraining LMs. Due to resource constraints, we randomly sample 235 million documents from this dataset, amounting to 306 billion tokens in total. We use the same pretraining data for all models.

**Model Details.** We take the model architecture from LLaMA (Touvron et al., 2023a) and train models across various sizes: 0.3, 0.7, 1.5, and 7.0 billion parameters, all with an 8192-length context window. Following LLaMA, we employ the AdamW optimizer (Loshchilov & Hutter, 2018) with parameters $\beta_1 = 0.9$ and $\beta_2 = 0.95$, and a cosine learning rate schedule. The 7B model is pretrained using 128 A100 GPUs across 16 nodes with a batch size of 4 million tokens. It takes 9 days to train the 7B model on our pretraining dataset. Due to the long context window of our models, we use flash attention (Dao et al., 2022) to reduce memory consumption during pretraining.

To perform the retrieval over our pretraining datasets, we employ the contriever model (Izacard et al., 2022) and encode the first 512 token of each document into an embedding. We then construct FAISS big batch search that is designed for conducting efficient similarity search with big batches of vectors (typically 50M–100M vectors per batch). Given each query document, we retrieve top 10 documents ($k$=10). We split the data in batches of 50M embeddings, the search step is conducted in each batch before merging the results using 8 GPUs per batch. The total search time is 6 hours over 32 GPUs with average search time per batch is 4,738s. The document graph traversal phase requires 12 hours on a setup of 20 CPUs.

More details are provided in the Appendix A.2.

## 3.2 BASELINES

We compare IN-CONTEXT PRETRAINING with the following baselines: (1) *Standard* is the prior standard in pretraining that places randomly shuffled documents in the input contexts. This method is commonly adopted by existing models (Zhang et al., 2022; Scao et al., 2022; Touvron et al., 2023a). (2) $k$NN (also referred to as retrieval-augmented language model pretraining (Guu et al., 2020; Levine et al., 2022)) directly places each document and its retrieved top-$k$ documents together in the same input context. Given the same number of training steps, $k$NN exposes LMs to a less diverse set of documents, since documents can repeat. For fair comparison, both standard and $k$NN methods are trained using the same pretraining data as IN-CONTEXT PRETRAINING and undergo an identical number of training steps, ensuring the same computation cost.

## 3.3 RESULTS

We perform evaluations on tasks that require understanding of contexts including language modeling (§ 3.3.1), in-context learning (§ 3.3.2), reading comprehension (§ 3.3.3) and open-book question answering (§ 3.3.4), factuality (§ 3.3.5) and long context reasoning (§ 3.3.6).

### 3.3.1 LANGUAGE MODELING

**Datasets & Metrics.** We evaluate the language modeling perplexity of IN-CONTEXT PRETRAINING and baselines on the Wikipedia, Arxiv, and Books corpora. We follow the standard language modeling evaluation in concatenating randomly-ordered documents when computing perplexity.

**Results.** Figure 3 shows average perplexity across different model sizes. First, $k$NN does not improve over the standard LM, likely due to the overfitting problem as discussed in §2.2. ICLM, in contrast, outperforms both the standard LM and $k$NN on all three datasets, even when the evaluation documents are not sorted. The gains are consistent or larger as the size of the model scales. These improvements suggest that IN-CONTEXT PRETRAINING provides better pretraining signals, enabling LMs to better hone their language modeling abilities.

### 3.3.2 IN-CONTEXT LEARNING FOR TEXT CLASSIFICATION

**Datasets & Metrics.** In-context learning requires to perform a task without fine-tuning by conditioning on a few demonstration examples about the task. We evaluate the in-context learnig ability of ICLM using 32 demonstration examples. We use seven text classification datasets, including

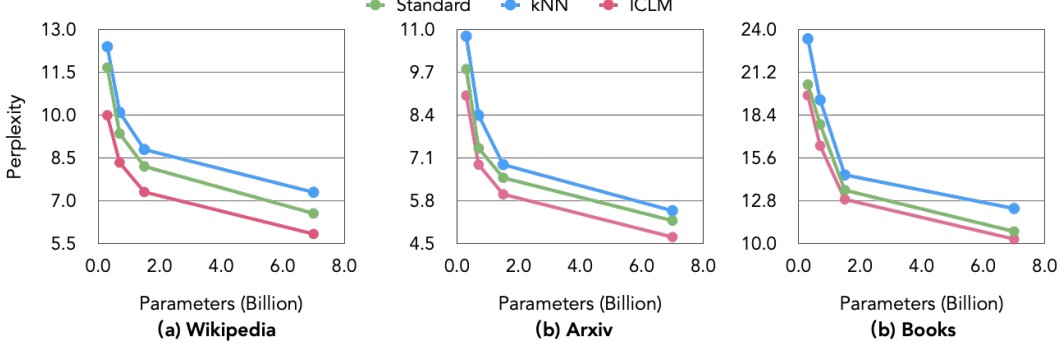

Figure 3: Language modeling perplexity (the lower the better) on Wikipedia, Arxiv, and Books (§3.3.1). ICLM **outperforms the baselines consistently across all model sizes.**

Table 1: In-context learning performance on seven classification datasets (§3.3.2). We use 32 in-context examples for all datasets. ICLM outperforms baselines on all datasets.

| Method | Sentiment | | | Hate Speech | | Topic Classification | | Average |
|---|---|---|---|---|---|---|---|---|
| | Amazon | SST2 | Yelp | Hate | Offensive | Agnews | Dbpedia | |
| Standard | 94.6 | 83.7 | 74.3 | 52.7 | 55.7 | 68.3 | 61.5 | 66.0 |
| kNN | 88.0 | 80.2 | 65.1 | 50.1 | 53.1 | 65.7 | 56.4 | 61.8 |
| ICLM | **96.5** | **93.2** | **77.4** | **60.6** | **57.3** | **76.0** | **63.2** | **71.3** |

sentiment analysis (SST-2 (Socher et al., 2013), Amazon and Yelp (Zhang et al., 2015a)), topic classificaiton (AGN (Zhang et al., 2015b) and Dbepdia (Lehmann et al., 2015)) and hate speech detection (Barbieri et al., 2020). We use label words from Min et al. (2022) and report accuracy as the metric.

**Results.** As shown in Table 1, ICLM consistently demonstrates better performance across all text classification datasets, leading to 8% gain on average. This result suggests that ICLM is better at learning from demonstration examples. We later analyze the relationship between the number of demonstration examples and the performance of the in-context learning in §4.3.

### 3.3.3 READING COMPREHENSION

**Datasets & Metrics.** Reading comprehension requires to answer the question based on the given paragraph. We consider the RACE reading comprehension benchmark (RACE-High and RACE-Middle) (Lai et al., 2017), SQuAD (Rajpurkar et al., 2016), BoolQ (Clark et al., 2019), DROP (Dua et al., 2019), and HotpotQA (Yang et al., 2018). We use 2-shot in-context learning for evaluation; we did not use more because some documents in reading comprehension tasks are very long. We report the exact match score for HotpotQA and SQuAD, and accuracy for other datasets that are multi-choice tasks (RACE, BoolQ, DROP), following the standard in prior work.

**Results.** Table 2 highlights that ICLM consistently surpasses both the standard and kNN baselines across all datasets with an average improvement of 14%. In particular, we observe significant gains on HotpotQA, which requires multi-hop understanding of multiple related documents. The performance gain on reading comprehension tasks demonstrates that IN-CONTEXT PRETRAINING improves LMs' ability of undestanding and reasoning over the given context.

### 3.3.4 RETRIEVAL-AUGMENTATION

**Datasets & Metrics.** Retrieval-augmentation is a method to retrieve a set of passages from the external text corpus (e.g., Wikipedia) and prepend it to the input query in order to better handle input queries that require factual knowledge (Lin et al., 2023; Xu et al., 2023; Su et al., 2023). We conduct evaluation on two well-studied open-domain QA datasets: Natural Questions (NQ) (Kwiatkowski et al., 2019) and TriviaQA (Joshi et al., 2017). For both datasets, we report exact match scores (EM) and evaluate the model performance in both closed-book and open-book settings. In the closed-book

Table 2: Reading comprehension results, using 2-shot in-context learning (§3.3.3). ICLM **outperforms baselines on all six datasets.**

| Method | RACE-High | RACE-Middle | BoolQ | SQuAD | HotpotQA | DROP | Average |
|--------|-----------|-------------|-------|-------|----------|------|---------|
| Standard | 39.5 | 53.3 | 68.9 | 26.3 | 10.5 | 27.2 | 37.6 |
| $k$NN | 36.2 | 51.4 | 65.3 | 23.5 | 14.4 | 25.1 | 36.0 |
| ICLM | **41.5** | **56.9** | **73.0** | **30.3** | **21.9** | **35.7** | **43.2** |

Table 3: Results on NQ and TQA (§3.3.4) without retrieval (closed) and with retrieval (open).

| Method | NQ | | TQA | |
|--------|--------|------|--------|------|
| | Closed | Open | Closed | Open |
| Standard | 17.0 | 28.5 | **49.3** | 48.1 |
| $k$NN | 13.5 | 20.1 | 40.2 | 43.2 |
| ICLM | 17.0 | **32.2** | 48.0 | **51.6** |

Table 4: Results on two datasets with knowledge conflicts, requiring better reasoning of the given context (§3.3.5).

| Method | NQ-Swap | MemoTrap |
|--------|---------|----------|
| Standard | 39.6 | 48.4 |
| $k$NN | 42.1 | 54.3 |
| ICLM | **45.8** | **56.2** |

setting, we only provide the question to the model and the model has to answer the question based on its parametric knowledge. In the open-book setting, we follow Shi et al. (2023c) in providing the model with the top-10 retrieved documents from Wikipedia as additional context to the question.

**Results.** Results are reported in Table 3. In the closed-book setting, ICLM performs comparably or slightly worse than the standard baseline, likely because our model memorizes less. Nonetheless, in the open-book setting, ICLM significantly outperforms the standard baseline in the open-book setting (+9%), obtaining much better performance than the closed book setting. It is also worth noting that the training objective of $k$NN is exactly the same as the retrieval-augmentation, but ICLM still achieves better performance, likely due to the overfitting problem of $k$NN as discussed in §2.2.

### 3.3.5 FACTUALITY

**Datasets & Metrics.** Prior work has found that language models generate text that is not factual nor faithful to the given context, especially when the context contradicts to knowledge the model has acquired during pretraining (often called parametric knowledge (Longpre et al., 2021; Zhou et al., 2023; Shi et al., 2023b; Wang et al., 2023)). We evaluate LMs' ablities to follow instructions and contexts on two knowledge conflict datasets: NQ-Swap (Longpre et al., 2021) and MemoTrap (Liu & Liu, 2023). Both datasets contain instruction and contexts that are in conflict with the models' parametric knowledge. We report exact match score as the metric.

**Results.** Table 4 shows that ICLM is better than the standard and $k$NN baselines on both datasets, implying that IN-CONTEXT PRETRAINING improves LMs' ability to generate outputs that are faithful to prior contexts. Gains are larger than those in other datasets, likely because NQ-Swap and MemoTrap highlight the challenge in reasoning about the given context, which the previous LMs struggle with.

### 3.3.6 LONG CONTEXT REASONING

**Datasets & Metrics.** To evaluate the ability of long context reasoning, we compare ICLM with the standard and $k$NN baselines on the SCROLL benchmark (Shaham et al., 2022) that evaluates LMs' ability to synthesize information over long texts. Following the original paper setting, we finetune the pretrained LMs (standard, $k$NN, IN-CONTEXT PRETRAINING) on the training datasets of the scroll and evaluate them on the test datasets. We report $F1$ score for Narrative QA, Qasper and ContractNLI datasets and report $ROUGE$-1 score for QMSum and GovReport datasets in the SCROLL benchmark.

**Results.** Results in Table 5 show that ICLM outperforms the baselines by around 5%, suggesting that ICLM is better at long context reasoning. We hypothesize that the gains from ICLM may fade

Table 5: Performance on long context reasoning benchmarks from SCROLL (Shaham et al., 2022) (§3.3.6). ICLM **outperforms baselines on all five datasets.**

| Method | NarrativeQA | Qasper | ContractNLI | QMSum | GovReport | Average |
|---|---|---|---|---|---|---|
| | *F1* | | | *ROUGE-1* | | |
| Standard | 16.5 | 34.2 | 78.6 | 25.1 | 8.2 | 32.5 |
| *k*NN | 16.8 | 34.1 | 79.5 | 24.3 | 6.6 | 32.3 |
| ICLM | **17.1** | **36.7** | **80.7** | **26.8** | **9.1** | **34.1** |

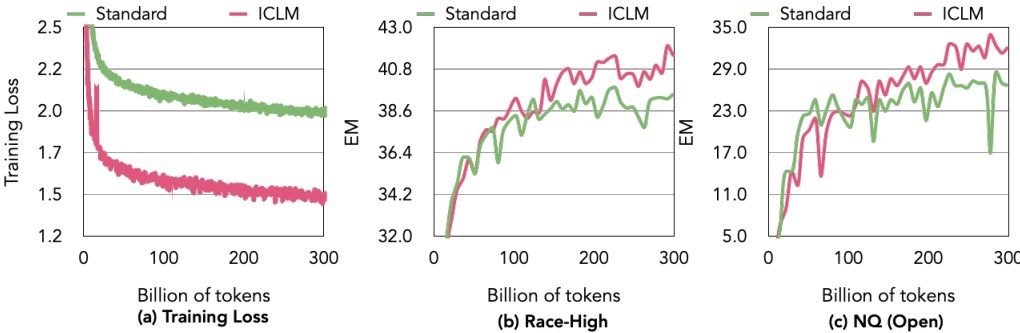

Figure 4: Training loss and performance evolution on reading comprehension during pretraining. **After training on around 150 billion tokens, ICLM is consistently better than the standard LM on reading comprehension and retrieval augmentation tasks.**

| Method Design | Choice | PPL |
|---|---|---|
| Document Relevance | Random | 8.2 |
| | Clustering | 7.9 |
| | Links (*final*) | 7.3 |
| Semantic Dedup | No dedup | 8.3 |
| | Dedup (*final*) | 7.3 |

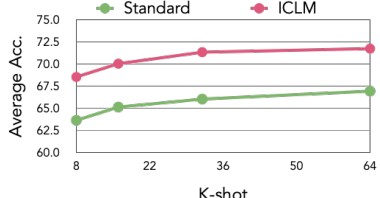

Figure 5: Ablation study of our method design.

Figure 6: Performance with respect to the number of in-context examples ($k$).

out to some extent when the LMs are fine-tuned, which may explain the relatively small gains in this evaluation compared to our other experiments.

# 4 ANALYSIS

## 4.1 EVOLUTION OF PERFORMANCE DURING PRETRAINING

Throughout the pretraining process, we closely monitor both the training loss and the downstream task performance for the ICLM as well as the standard LM. Figure 4 illustrates the trajectory of the training loss and the performance on the RACE reading comprehension tasks for the 7B models. The training loss for ICLM consistently remains lower than that of the standard LM. This suggests that, when predicting the next token, ICLM benefits from a richer set of relevant prior documents to refer to, while the standard LM has limited information to rely on, leading to higher loss. Figure 4 (b, c) shows that after training on around 150 billion tokens, ICLM is consistently better than the standard LM on reading comprehension tasks. This performance gap remains consistent throughout the remainder of the pretraining phase. This suggests the scale of improvements by IN-CONTEXT PRETRAINING does not diminish and remains consistent as training on more tokens.

## 4.2 Ablation Study on In-Context Pretraining Design

We perform analysis on two design choices of IN-CONTEXT PRETRAINING: a choice of methods for finding retrieved documents and deduplication. Ablations are done with 1.5B models and evaluated with perplexity on Wikipedia. The results are presented in Figure 5.

**Document relevance.**   A key design of IN-CONTEXT PRETRAINING is grouping documents by their relevance. We consider three levels of relevance: random (the standard baseline discussed in §3.2), clustering, and our document linking method in IN-CONTEXT PRETRAINING. Clustering follows the method from Abbas et al. (2023) in clustering documents into 11k clusters based on their embeddings and sample documents from each cluster to form the training inputs. Documents grouped by clustering are sourced from the same clusters, indicating topical similarity but not necessarily close relation. In contrast, ICLM links documents as nearest neighbors, indicating a higher degree of similarity. The relevance between documents increases from random, clustering to linking. We observe that the perplexity of the language model decreases as the relevance increases.

**Deduplication.**   We compare perplexity of the models trained with and without the semantic deduplication step. Removing the semantic deduplication step leads to a significant decrease in perplexity. When near duplicate documents are present in the same context, language models might merely copy from the prior document, leading to training instability.

## 4.3 Demonstration examples size for in-context learning

We evaluate the 7B models trained with the standard method and IN-CONTEXT PRETRAINING, using a varying number of demonstration examples on text classification tasks described in §3.3.2. As depicted in Figure 6, ICLM maintains consistent performance gains over the standard method, even as the number of demonstration examples grows. While the performance improves as the number of demonstration examples increases, it plateaus after 32 examples.

## 5 Related Work

**Data batching based on similarity**   Previous work employs batching lexically similar segments in the same training batches to construct high-quality positive pairs for training retrieval-augmented language models. For instance, Zhong et al. (2022) use BM25 and same documents to ensure the segments in the same batch are similar to each other, while Min et al. (2023) group segments from the same documents in the same batch. Our method shares the same spirit with these methods except we maintain the relevance of documents in the same context window, yet context windows within batches are shuffled. Additionally, our focus is to apply the batching method to train the standard language models.

## 6 Conclusion

We introduce IN-CONTEXT PRETRAINING, a new pretraining method that learns to generate text conditioned on a set of relevant documents, exposing LMs to relevant contexts and providing training signals beyond document boundaries. Our method is highly scalable and simple, and works with any pre-training pipeline by simply changing the document ordering during preprocessing. Our comprehensive evaluation demonstrates our method leads to significant improvements in a wide variety of settings that highlight the ability to understand and reason over the given context, including in-context learning, reading comprehension, retrieval augmentation, and more. Future research may delve into the inherent connections between documents within specific corpus domains or using multilingual retriever to group related multilingual documents in the same context. For example, the code scripts within the same repository are related. This insight paves the way for future exploration, where concatenating entire repositories into a unified whole could lead to the creation of meaningful long-context data sets.

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

# A  ADDITIONAL BACKGROUND

## A.1  DEDUPLICATION

Corpora often have semantic duplicates: pairs of documents that are semantically related, yet not entirely identical. Previous studies (Yasunaga et al., 2023) show that retraining highly similar documents in the input contexts during training hurts multimodal models' performance. We observed a similar behaviur: when near duplicate documents are present in the same context, language models might merely copy from the prior document, leading to training instability. Given that our retrieval approach inherently assesses pairwise document similarity, we can easily filter out near duplicate documents that have high cosine similarity with other documents. We find this deduplication step to be crucial for achieving performance of good language models (§4.2).

## A.2  FAISS INDEX

We used a product quantised inverted file (IVFPQ) FAISS index with a code size of 256 and the corresponding number of inverted lists 32768 , with total size of 62 gigabytes. The index contains 235266464 768-dimensional embeddings originally in float 32. The index was trained on a sample of 1572864 embeddings and the train time was 423 s. Successively, the data is split in batches of 50M embeddings and for each index shard the corresponding batch of embeddings is added to the trained index, the average adding embeddings time per index shard is 628.4 s. Finally, approximate nearest neighbor search is conducted per each shard before aggregating all results using faiss big batch search. The nprobe used for conducting approximate search is 64, this means that 0.2% of the inverted lists are probed during the nearest neighbors search.

