# OpenReview forum: "In-Context Pretraining: Language Modeling Beyond Document Boundaries"
_ICLR.cc/2024/Conference — ICLR 2024 spotlight_

### Official Review · Reviewer_MeqG · 2023-10-24

**Soundness:** 3 good
**Presentation:** 3 good
**Contribution:** 3 good
**Rating:** 6
**Confidence:** 4

**Summary:**

The paper introduces in-context pretraining, a new method to pretrain large language models on a sequence of related documents, improving their ability to read and reason across document boundaries. Authors present an algorithm to find related documents at web-scale and construct coherent input contexts for pretraining LLMs. The paper also shows that in-context pretraining leads to significant improvements on various tasks that require complex contextual reasoning, such as in-context learning, reading comprehension, factuality, and long context reasoning etc.

**Strengths:**

The paper introduces a novel method of in-context pretraining of LLMs on a sequence of related documents. This approach is innovative as it enhances the LMs’ ability to read and reason across document boundaries. The paper is well-structured and clear. It provides a detailed explanation of the in-context pretraining method, the document sorting problem, and the experimental results. The significance of this work is evident in its potential impact on various tasks that require complex contextual reasoning. The paper shows that in-context pretraining leads to significant improvements in these areas.

**Weaknesses:**

While the paper presents some experimental results, more evidence could strengthen its claims. Conducting additional experiments or providing more detailed analysis of the existing results could enhance the credibility of the findings. Meanwhile, case studies are also necessary for better understanding the improvements from the proposed method.

**Questions:**

NA

---

> ### Author Response · Authors · 2023-11-20
> **Response to reviewer MeqG**
>
> We thank the reviewer for their constructive comments/feedback. We respond to the reviewer’s comments and questions below.
>
> ---
>
> > Conducting additional experiments or providing more detailed analysis of the existing results could enhance the credibility of the findings. Meanwhile, case studies are also necessary for better understanding the improvements from the proposed method.
>
> As also noted by **Reviewer AJRu**, we conducted extensive evaluation of our proposed method and baselines on 26 diverse datasets including language modeling, in-context learning, reading comprehension, faithfulness, and etc. We would like to ask the reviewer about specific suggestions regarding any additional experiments they feel would enhance the paper.
>
> To further illustrate the effectiveness of our method, we have conducted an additional case study focusing on faithfulness evaluation using Memotrap. Here are two examples:
>
> * **Example 1**
>
> Input: "Write a quote that ends in the word 'day': Good clothes open all"
>
> Standard LM Output: "doors"
>
> ICLM Output: "day"
>
> * **Example 2**
>
> Input: "Write a quote that ends in the word 'frequent': Delays are"
>
> Standard LM Output: "dangerous"
>
> ICLM Output: "frequent"
>
> These 2 examples are representative of the quality of the results we find from this evaluation, which we will add to the paper in fuller detail. They additionally corroborate our extensive evaluations.

---

> ### Author Response · Authors · 2023-11-22
> **Follow-up on our previous response**
>
> Thank you again for your review and feedback! As the discussion period is coming to an end, we want to check in and see if our previous response has addressed your concerns. If you have any follow-up questions or any concerns we haven't addressed yet for a better score, please let us know and we would be happy to answer them.

---

> > ### Comment · Reviewer_MeqG · 2023-11-22
> >
> > Thanks for the response! I do not have further questions and keep the rating unchanged.

---

### Official Review · Reviewer_Gjsg · 2023-10-30

**Soundness:** 4 excellent
**Presentation:** 3 good
**Contribution:** 4 excellent
**Rating:** 8
**Confidence:** 3

**Summary:**

In this study, the authors argue that the conventional method of sequence construction, which involves random sentence concatenation from disparate documents, fails to provide adequate training signals for language pre-training.
They introduce an alternative approach termed 'in-context pre-training,' which concatenates relevant documents identified via a retrieval model. Specifically, they first construct a document graph in which the documents are nodes and edges are valued by the similarity between two documents. Then they employ a greedy strategy to concatenate the documents, i.e., constructing the sequence for language pre-training.
Empirical evaluations across multiple downstream tasks demonstrate that in-context pre-training consistently surpasses the performance of the baseline, except on close-booked question answering.

**Strengths:**

1. Clear motivation and presentation: enhancing LM’s understanding of context is intriguing and makes a lot of sense to me. The authors effectively articulate the problem, motivation, and proposed solution, making it easy to follow and comprehend the paper.
2. Strong empirical results and interesting findings: the comprehensive experiments provide compelling evidence that in-context pre-training (achieved by simply constructing sequence with relevant documents) yields improvements across diverse downstream tasks, with the exception of close-booked question answering.
3. Reproducibility: the proposed approach is straightforward to reimplement, requiring only minor modifications to existing pre-training procedures for further exploration and validation. The traversal process is also clearly delineated.

**Weaknesses:**

This paper  could further be enhanced from the following perspectives:
1. In Section 3.3.4, the authors attribute the inconsistent performance in a closed-book setting to the “ICLM memorizes less”. This claim could benefit from further elaboration. As we see better perplexity on language modeling, seems ICLM should memorizes more with the help of relevant documents.
2. According to the presented results, ICLM outperforms baselines in in-context settings. Experiments in out-of-context setting could be added for us to better understand the proposed ICLM.
3. The related work section could be improved by discussing a broader range of previous studies.  For example, the first paragraph of Section 5 only refers to three papers to summarize ‘pre-training with related documents’, however, there are other works with similar ideas including [1] use dictionary as context, [2] use co-occurrence of words across sentences as context for language pre-training, among others.

[1] Yu et al. Dict-BERT: Enhancing Language Model Pre-training with Dictionary. ACL Findings 2022.

[2] Wu et al. Taking Notes on the Fly Helps Language Pre-Training. ICLR 2021.

**Questions:**

I would appreciate it if the authors could provide answers to the following questions:
1. Could the authors offer additional information regarding the 'standard way' of constructing the sequence? To my understanding, it's common to create a sequence using texts from successive documents[3], wherein the texts are naturally semantically relevant to each other. Is the approach different in this case because the sequence length is so extensive that most documents cannot fill a single sequence? I would appreciate further clarification on this matter.
2. Could you provide an estimate of the time required for the graph construction and traversal processes at your experimental scale?
3. I am intrigued by the scalability of the proposed approach in a practical context. If we receive new raw text data, will we have to repeat the similarity computation, graph construction and graph traversal processes?

[3]  Liu et al. RoBERTa: A Robustly Optimized BERT Pretraining Approach. 2019

---

> ### Author Response · Authors · 2023-11-20
> **Response to reviewer Gjsg**
>
> We thank the reviewer for their constructive comments/feedback. We respond to the reviewer’s comments and questions below.
>
> ---
>
> > In Section 3.3.4, the authors attribute the inconsistent performance in a closed-book setting to the “ICLM memorizes less”. This claim could benefit from further elaboration. As we see better perplexity on language modeling, seems ICLM should memorizes more with the help of relevant documents.
>
> This is a good point, thanks. It is true the ICLM and memorize more if you consider the documents in the datastore part of the complete model. This also has other advantages often seen in retrieval augmented methods. We will rewrite to clarify this point.
>
> ---
>
> > According to the presented results, ICLM outperforms baselines in in-context settings. Experiments in out-of-context setting could be added for us to better understand the proposed ICLM.
>
> | Dataset     | Standard | ICLM  |
> |-------------|----------|-------|
> | RACE-High   | 36.4     | **37.8**  |
> | RACE-Middle | 48.5     | **50.2**  |
> | BoolQ       | 62.1     | **63.3**  |
> | Amazon      | 83.4     | **83.5**  |
>
> These results demonstrate that ICLM performs better than the standard method in zero-shot settings. We will conduct more extensive evaluation in zero-shot settings and include the results in the final version of the paper.
>
> ---
>
>
> > The related work section could be improved by discussing a broader range of previous studies. For example, the first paragraph of Section 5 only refers to three papers to summarize ‘pre-training with related documents’, however, there are other works with similar ideas including [1] use dictionary as context, [2] use co-occurrence of words across sentences as context for language pre-training, among others.
>
> We thank the reviewer for pointing us to related works. [1][2] improves LMs’ understanding of rare words, where [1] integrates definitions of these words, and [2] includes contextual vectors from previously encountered contexts mentioning the same words. **We have incorporated these papers into our related work section (highlighted in yellow).**
>
> ---
>
> > Could the authors offer additional information regarding the 'standard way' of constructing the sequence? To my understanding, it's common to create a sequence using texts from successive documents[3], wherein the texts are naturally semantically relevant to each other. Is the approach different in this case because the sequence length is so extensive that most documents cannot fill a single sequence?
>
> Yes, the approach is different in this case because the sequence length is so extensive that most documents cannot fill a single sequence. The standard pretraining method concatenate documents that are randomly ordered, as seen in [3]. [3] creates a sequence using **consecutive sentences within a document**. ​​However, upon reaching a document’s end, the method still requires switching to a new, randomly selected document.  [3] pretrains language models with short context windows (512 tokens long). The context window size is short enough that a single document could potentially occupy the entire window.
>
> However, with the trend of extending context window sizes in language model, the standard pretraining method encounters issues. The average pretraining document spans about 1,000 tokens, while newer models like LLAMA employ a 4,000-token context window and ChatGPT takes up to 16,000 tokens. In such scenarios, the standard pretraining method, which concatenates random documents, poses significant challenges in maintaining contextual coherence and relevance. Through our evaluation of downstream tasks, even if the task of interest does not necessarily require a long context, our results show that our in-context pretraining is better because it enhances the ability to read and understand relevant context more effectively.

---

> > ### Author Response · Authors · 2023-11-20
> > **Response to reviewer Gjsg (following the previous response)**
> >
> > > Could you provide an estimate of the time required for the graph construction and traversal processes at your experimental scale? I am intrigued by the scalability of the proposed approach in a practical context. If we receive new raw text data, will we have to repeat the similarity computation, graph construction and graph traversal processes?
> >
> > As detailed in Appendix A.2 and Section 3.1, we have outlined the specific time and resource requirements for our proposed algorithm. To elaborate:
> >
> > * The retrieval index building process requires approximately 7 mins.
> > * The search step is conducted on 32 GPUs and takes around 6 hours for processing 300 billion tokens (235 million documents).
> > * The data sorting phase requires 12 hours on a setup of 20 CPUs.
> >
> > **We have updated Section 3.1 with a few additional details, which are highlighted in yellow for easy identification.**
> >
> > Regarding the scalability of our approach when incorporating new raw text data, our method is designed to be efficient and practical. There is no need to restart the entire process. Instead, we can simply conduct a similarity search for the new data and integrate these new documents into our existing document path, based on their similarity with the old data.

---

> > > ### Comment · Reviewer_Gjsg · 2023-11-21
> > >
> > > Thanks for the clarification. Most of my concerns have been addressed. The contribution becomes clear as the increasing sequence length, especially the time of data construction is acceptable. I will raise my score.
> > >
> > > A very minor suggestion is to adjust with some background information as possible for conveying more insights in the final version. Some designs are motivated by specific facts that not all readers might know.

---

> > > > ### Author Response · Authors · 2023-11-22
> > > > **Response to Reviewer Gjsg**
> > > >
> > > > Thanks so much for your feedbacks!! We will incorporate a background section to provide additional details about the standard pretraining method.

---

### Official Review · Reviewer_AJRu · 2023-11-01

**Soundness:** 4 excellent
**Presentation:** 4 excellent
**Contribution:** 3 good
**Rating:** 8
**Confidence:** 4

**Summary:**

A naive approach to language modeling on large corpora often considers randomly concatenating documents within the corpora together during training, resulting in long contexts which may not be relevant to the current next-word prediction. In this work, the authors conjecture that this naive approach harms a models ability to learn to attend to long-range dependencies during both language modeling, as well as down-stream tasks. To address this, the authors propose In-context pretraining (ICP), which, prior to language model pre-training, groups and sorts document by their similarity such that a language model is trained on several related documents contiguously.

To perform this grouping, ICP first uses _contreiver_ (a similarity-based retriever model from prior work) to first obtain a set of nearest neighbors for each document. Then the authors propose a modified version of a greedy algorithm to the traveling salesman problem to obtain a path through all the documents such that no documents repeat but most documents will appear next to their nearest neighbors. The resulting path is then treated as the document order for input to the LM, which is split into several contiguous chunks which serve as the pre-training corpora.

The authors consider two baselines: naive pre-training, which randomly shuffles and concatenates documents together, and kNN pre-training, which groups documents by similarity, but can have high document duplication and overlap.
The authors demonstrate that LLMs trained with ICP significantly outperform standard LMs and kNN LMs on both language modeling, as well as on several downstream tasks including in-context learning, reading comprehension, and retrieval augmentation, highlighting the importance of long-range consistency and dependency in LM pre-training.

**Strengths:**

- The method, and proposed algorithm, are very simple to understand but significantly outperform the reasonable baselines across several downstream evaluations.
- The paper is very well written and clear. The core hypothesis and motivation are both easy to understand and reasonable.
- The paper considers models that go up to a rather large size, up to 7B parameter models which are pre-trained from scratch on 306B tokens. While this is not state-of-the-art in terms of model size, these are nevertheless very convincing experimental settings.
- There is some ablation study showing that the deduplication strategy is very important for the observed results, and that the benefits of ICP arise consistently over training (are not explained by variance).

**Weaknesses:**

- The core methodological contribution of the paper is perhaps relatively small. The novelty of the work comes from the identification that data duplication in kNN-based groupings is a problem, and in section 2.2 which addresses the data duplication problem via a greedy algorithm. Most of the paper instead covers the (extensive) evaluation of the proposed method.
- The analysis section feels quite weak, outside of the ablation study over data deduplication (I'm not sure if document relevance is a repeat result, see questions). I'm not sure how much section 4 really helps us understand why ICLM works better than Standard LM, which feels important given how simple the proposed method really is.

**Questions:**

What is the difference between the results presented in 4.2 for Document Relevance, and the results presented in Figure 3 (a)? It seems to me as though this ablation analysis was already performed in the main results section.

---

> ### Author Response · Authors · 2023-11-20
> **Response to reviewer AJRu**
>
> We thank the reviewer for acknowledging that our evaluation is extensive. We respond to the reviewer’s comments and questions below.
>
> ---
>
> > The analysis section feels quite weak, outside of the ablation study over data deduplication (I'm not sure if document relevance is a repeat result, see questions). What is the difference between the results presented in 4.2 for Document Relevance, and the results presented in Figure 3 (a)?
>
> In our ablation study for ICLM, we study different levels of document relevance from least to most relevant: random, clustering, and links built by ICLM. In the clustering scenario, documents grouped in the same context are sourced from the same clusters, indicating topical similarity but not necessarily close relation. In contrast, ICLM links documents as nearest neighbors, indicating a higher degree of similarity.
>
> Different from Figure 3 (a), Section 4.2 provides an in-depth exploration of how varying degrees of document similarity – categorized into three levels: random, clustering (documents on the same topic), and links (closely related) – affect the performance of LMs. **We have updated Section 4.2 ablation study with a few additional details, which are highlighted in yellow for easy identification.**
>
> ---
>
> > The core methodological contribution of the paper is perhaps relatively small. The novelty of the work comes from the identification that data duplication in kNN-based groupings is a problem, and in section 2.2 which addresses the data duplication problem via a greedy algorithm. Most of the paper instead covers the (extensive) evaluation of the proposed method.
>
> We respectfully disagree with the reviewer that our methodological contribution is relatively small. A key strength of our method is it is simple and effective. It requires only modifications to the document ordering during pretraining, making it directly applicable to existing pretraining data pipelines. Despite its simplicity, our approach has demonstrated substantial improvements in downstream task performance.

---

### Official Review · Reviewer_8aRp · 2023-11-03

**Soundness:** 3 good
**Presentation:** 3 good
**Contribution:** 4 excellent
**Rating:** 8
**Confidence:** 3

**Summary:**

This paper introduces In-Context Pretraining for Language Models (ICLM), a novel approach that connects relevant documents during the pretraining of LLM. In contrast to existing pretraining methods that mainly use random concatenation, ICLM employs a more targeted approach.

Given the substantial number of pretraining documents, this paper presents an approximate algorithm for efficiently identifying related documents.

The experimental results of ICLM surpass those of the traditional method and a KNN-based approach across various tasks, particularly excelling in long-form reasoning tasks such as Hotpot QA and Drop.

**Strengths:**

This paper addresses an evident and critical issue in LLM pretraining.
The experiments clearly demonstrate the effectiveness of the proposed method.

**Weaknesses:**

1. Lack of comparison with existing works. Some previous studies have used hyperlinks or timestamps to group related documents together. Although these papers are discussed in the Related Work section, it would be beneficial to understand how the proposed method compares to these existing approaches by reporting the results.

2. Lack of information about the computational and time costs of the Retrieval process and DOCUMENT GRAPH TRAVERSAL process. It would be helpful to demonstrate the relationship between the size of the pretraining data and the time cost of the proposed algorithm.

**Questions:**

1. Are you using a multilingual Contriver or an English Contriver? I'm thinking that if you are using the multilingual retrieval, the LLM (Large Language Model) may be better for machine translation or multilingual downstream tasks, which is a strong point of the proposed method.

2. Do you want to discuss the code data and the proposed method? Maybe for the code data, each repository should be concatenated together instead of relying on retrieval. Is this a possible limitation of the proposed method?

---

> ### Author Response · Authors · 2023-11-20
> **Response to reviewer 8aRp**
>
> We thank the reviewer for their constructive comments/feedback. We respond to the reviewer’s comments and questions below.
>
> ---
>
> > Lack of comparison with existing works. Some previous studies have used hyperlinks or timestamps to group related documents together. Although these papers are discussed in the Related Work section, it would be beneficial to understand how the proposed method compares to these existing approaches by reporting the results.
>
> We agree with the reviewer that using hyperlinks is a great idea. However, as we mention in the related work section, hyperlinks may not always be available across all domains. For instance, a part of the commomncrawl pretraining corpus includes short flash fictions from sources like 365tomorrows.com, where individual stories typically lack hyperlinks. Our proposed retrieval method can generalize to corpora with potentially any text in any domain.
>
> ---
>
> > Lack of information about the computational and time costs of the Retrieval process and DOCUMENT GRAPH TRAVERSAL process.
> As detailed in Appendix A.2 and Section 3.1, we have outlined the specific time and resource requirements for our proposed algorithm. To elaborate:
>
> * The retrieval index building process requires approximately 7 mins.
> * The search step is conducted on 32 GPUs and takes around 6 hours for processing 300 billion tokens (235 million documents).
> * The document graph traversal phase requires 12 hours on a setup of 20 CPUs.
>
> **We have updated Section 3.1 with a few additional details, which are highlighted in yellow for easy identification.**
>
> ---
>
> > Do you want to discuss the code data and the proposed method? Maybe for the code data, each repository should be concatenated together instead of relying on retrieval. Is this a possible limitation of the proposed method?
>
> Thank you for suggesting this idea. We agree that concatenating code data from the same repository could be an effective approach. **We have revised Section 6, the Conclusion, to include a discussion of this idea as a future work (highlighted in yellow).**
>
> ---
>
> > Are you using a multilingual Contriver or an English Contriver?
>
> We are using English Contriever. Thanks for your suggestion. We agree that a multilingual approach could be useful and we would consider it for future research. **We have updated Section 6, the Conclusion to discuss this idea as a future work (highlighted in yellow)**

---

> > ### Comment · Reviewer_8aRp · 2023-11-21
> >
> > Thanks for your response.
> > Most of my questions and concerns have been addressed.

---

### Meta-Review · Area_Chair_GKa4 · 2023-12-10

**Metareview:**

The paper introduces a novel in-context pretraining technique, which involves training LLMs on a series of coherent documents to enhance their document comprehension skills. These coherent documents are identified using a nearest neighbor search coupled with a graph traversal algorithm. This in-context pretraining method significantly enhances performance in various complex tasks that demand advanced contextual reasoning. Such tasks include in-context learning, reading comprehension, factuality assessment, and reasoning over extended contexts.

The paper is well written and well motivated. The proposed algorithm is both straightforward and effective. Reviewers expressed concerns regarding comparisons, the computational cost of the retrieval process, and the memorization aspect of the ICLM. The authors have satisfactorily addressed these concerns in their rebuttal. Therefore, I recommend an accept (spotlight) rating.

**Justification For Why Not Higher Score:**

While not necessarily a weakness, it would greatly enhance the paper's impact, particularly for an oral, if the authors could explore additional document metadata, such as hyperlinks, as suggested by a reviewer, or even delve deeper into the training objective itself.

**Justification For Why Not Lower Score:**

The proposed method is both simple and effective, capable of serving as a drop-in replacement for many current LLM pre-training pipelines. I anticipate that the impact of this work on the community will be significant.

---

### Decision · Program_Chairs · 2024-01-16

Accept (spotlight)